# Rare diseases load through the study of a regional population

**Élisa Michel**[1,2], **Claudia Moreau**[1,2], **Laurence Gagnon**[1,2], **Mylène Gagnon**[1,2], **Josianne Leblanc**[3], **Jessica Tardif**[3], **Lysanne Girard**[4], **Jean Mathieu**[5,6], **Cynthia Gagnon**[5,6,7], **Mathieu Desmeules**[5,6,8], **Jean-Denis Brisson**[5,6,9], **Luigi Bouchard**[3,4,7], **Simon L. Girard**[1,2,10,11] *

**1** Département des sciences fondamentales, Université du Québec à Chicoutimi, Saguenay, Québec, Canada, **2** Centre Intersectoriel en Santé Durable (CISD), Université du Québec à Chicoutimi, Saguenay, Québec, Canada, **3** Département Clinique de Médecine de Laboratoire du Centre Intégré Universitaire de Santé et Services Sociaux (CIUSSS) du Saguenay–Lac-St-Jean, Saguenay, Québec, Canada, **4** Département de Biochimie et de Génomique Fonctionnelle, Faculté de Médecine et des Sciences de la Santé, Université de Sherbrooke, Saguenay, Québec, Canada, **5** Groupe de Recherche Interdisciplinaire sur les Maladies Neuromusculaires (GRIMN), CIUSSS du Saguenay–Lac-Saint-Jean, Saguenay, Québec, Canada, **6** Faculté de Médecine et des Sciences de la santé, Université de Sherbrooke, Saguenay, Québec, Canada, **7** Centre de Recherche et d'innovation du CIUSSS du Saguenay–Lac-St-Jean, Saguenay, Québec, Canada, **8** Clinique de pédiatrie du Saguenay, Saguenay, Québec, Canada, **9** Clinique des Maladies Neuromusculaires (CMNM), CIUSSS du Saguenay–Lac-St-Jean, Saguenay, Québec, Canada, **10** Projet BALSAC, Université du Québec à Chicoutimi, Saguenay, Québec, Canada, **11** Centre de recherche CERVO, Université Laval, Québec, Canada

* simon2_girard@uqac.ca

## Abstract

Rare genetic diseases impact many people worldwide and are challenging to diagnose. In this study, we introduce a novel regional population cohort approach to identify pathogenic variants causing Mendelian diseases that occur more frequently within specific populations and are of clinical interest for carrier testing. We utilized a cohort from Quebec, including the Saguenay–Lac-Saint-Jean region, which is known for its founder effect followed by a rapid expansion and higher frequency of certain pathogenic variants. By analyzing both their frequency and origin through shared identical-by-descent segments, we identified founder variants. We calculated and compared their frequency in individuals originating from the Saguenay–Lac-Saint-Jean and from other urban Quebec regions. We validated 38 previously reported variants as being more common due to the founder effect and population expansion. Additionally, we identified 42 unreported founder variants in Quebec or Saguenay–Lac-Saint-Jean, some with carrier rates estimates as high as 1/22. We also observed a greater deleterious mutational load for the studied variants in individuals from the Saguenay–Lac-Saint-Jean compared to other urban Quebec regions. These findings were brought to the clinic, where 12 pathogenic variants were detected in diagnosed patients. Five variants found in this study are responsible for very severe diseases and could be considered for inclusion in a carrier test for the Saguenay–Lac-Saint-Jean population. This study highlights the potential underestimation of rare disease



**Data availability statement:** Quebec genotype, imputed and WGS data are available under restricted access from CARTaGENE biobank (https://cartagene.qc.ca/en/researchers/access-request.html) due to the informed consent given by study participants. The code used and data for this study can be found in the following GitHub repository: https://github.com/Genopop/Figures-founder-variants-article.

**Funding:** Funding for SLG was provided by the Canada Research Chair in Genetics and Genealogy grant #CRC-2022-00444( https://www.chairs-chaires.gc.ca/chairholders-titulaires/profile-fra.aspx?profileId=5645). LB and CG were funded by the Research Chair in Génétique et parcours de vie en santé(https://www.chairegps.com/).The funders of the study had no role in study design, data collection, data analysis, data interpretation, or writing of the article.

**Competing interests:** The authors have declared that no competing interests exist.

prevalence and presents a population-based approach that could aid clinicians in their diagnostic efforts and patients' management.

## Author summary

Rare genetic diseases present significant diagnostic challenges and impact individuals worldwide. In this study, we introduce an innovative regional population cohort approach to identify pathogenic variants that are more common in specific populations. We examined a cohort from Quebec, specifically the Saguenay–Lac-Saint-Jean region, known for its founder effect and subsequent population expansion. By analyzing both their frequency and recent origin, we identified 38 previously reported and 42 new founder variants, some with carrier rates as high as 1/22. Our analysis showed a higher deleterious mutational load in individuals from this region compared to other urban Quebec populations. Our findings were clinically validated, revealing 12 pathogenic variants in diagnosed patients. Five variants found in the present study are linked to severe diseases and could be incorporated into carrier screening for the Saguenay–Lac-Saint-Jean population. Our study underscores the importance of considering regional genetic variations in the diagnosis and management of rare diseases, offering a new framework for improving carrier testing and genetic counseling.

## Introduction

Rare diseases are thought to collectively affect as much as 10% of the population [1]. There are more than 6,000 rare Mendelian diseases described in Orphanet. Diagnosis remains a significant challenge for patients living with a rare disease. Despite the growing accessibility of genome sequencing technologies in precision medicine efforts for rare disease diagnosis [2], these patients often experience a prolonged diagnostic odyssey due to insufficient knowledge about their specific condition and the diversity of symptoms observed for a given disease. It becomes increasingly important to improve the diagnostic yield of rare diseases and to shorten the diagnostic odyssey of patients [3]. Understanding population health disparities is an essential component of equitable precision health efforts.

In certain populations, the prevalence of some rare diseases may increase due to demographic events such as founder effects and population expansions [4]. It is the case in Quebec, a province in Canada, predominantly settled by people of French origin starting in the early 1600s [5]. The initial European founder effect was followed by subsequent regional founder effects, notably the well-characterized one observed in the Charlevoix and Saguenay–Lac-Saint-Jean (SLSJ) regions [6]. This, followed by a very rapid population expansion, resulted in many rare diseases that are more frequent in SLSJ than elsewhere in the world [7–10]. In SLSJ, most people are aware of the higher risk of transmission of some rare diseases and a carrier test is offered to the populations of Charlevoix, SLSJ and Haute-Côte-Nord for six of these diseases

[7,8]. Nevertheless, numerous rare diseases still lack a known genetic etiology and diverse manifestations of diseases across patients further complicate clinical diagnosis. Traditionally, diseases with higher frequency in specific populations have been analyzed using a bottom-up approach, starting with the phenotypes of patients observed in clinical settings and linking them to genes or variants. Often, medical geneticists and the healthcare system would gain valuable insights from obtaining a comprehensive overview of variants that are more frequent in the population and potentially associated with rare diseases. This study focuses on addressing this need.

More specifically, we aimed to describe pathogenic variants that have an increased frequency in SLSJ due either to the founder effect and expansion or simply due to many introductions in the population. We conducted a comprehensive screening to identify pathogenic variants with higher frequency in SLSJ. Since the SLSJ population has been extensively studied over the past 40 years, we expected to identify many previously reported variants, thereby validating our findings. Importantly, the SLSJ healthcare system features a single entry point for all residents, which simplifies the process of locating patients with newly identified pathogenic variants.

Furthermore, we report for the first time the load of rare variants in a single population and assess how the founder effect and expansion were pivotal in increasing that load. In the context of rare diseases, a large number of populations remain poorly characterized, and we believe that our study highlights the need for regional genetic programs to better understand and diagnose the variety of rare diseases affecting populations.

## Results

### Rare pathogenic variants in the Quebec Province

We detected 9,043 rare pathogenic variants present either in individuals from the Quebec Province (QcP) or in gnomAD non-Finnish Europeans (nfe). 93% of these variants were present in gnomAD nfe, 16% in the QcP and only 10% in the SLSJ region (see methods for details on QcP regional clustering). Only two variants with a MAF greater than 0.005 are at least twice as frequent in the QcP than in gnomAD (in orange), whereas five such variants are at least twice as frequent in gnomAD compared to the QcP (in purple; Fig 1A). In contrast, 40 variants were more frequent in the SLSJ compared to gnomAD (Fig 1B). Considering only the variants present in the QcP, 25% are absent from the urban Quebec region (UQc; mean number of absent variants after 1,000 resamplings of 3,589 individuals to match the SLSJ sample size, see methods) and 43% are absent from the SLSJ (chi² p < 0,001). Unsurprisingly, we observed a lower proportion of individuals from the SLSJ that do not carry any pathogenic variant (chi² p < 0.001) compared to UQc resampling (Fig 2). Hence, there are fewer pathogenic variants in the QcP compared to gnomAD, and even fewer in the SLSJ region compared to UQc. However, there is a greater proportion of pathogenic variants at higher frequencies in the SLSJ than in UQc, QcP or gnomAD.

### Previously reported and newly discovered variants

Seventy-two variants were previously published in literature reviews on the Charlevoix-SLSJ founder effect [7–10]. Of these, 42 were present in our data (S1 Table). Table A in S1 Text provides details on the 30 previously reported variants that were either absent in our data or not considered in our analysis. Noticeably, there is a great correlation between the carrier rates (CR) previously reported and the ones calculated herein (Fig A in S1 Text). Moreover, nine carrier rates were assessed independently in the CIUSSS laboratory, and eight of the newly calculated rates do not differ (95% confidence interval) from those reported in this study (Table B and Supplementary Methods in S1 Text).

Based on the absence of a clear definition for founder variants, we propose here a definition: The variants must be more frequent in the QcP than in gnomAD (relative frequency difference (RFD) ≥ 10%, see methods), have a CR of at least 1/200 and at least 50% of the pairs of carriers must share a segment identical-by-descent (IBD) around the variant's position (see methods). Among the 1,302 rare pathogenic variants with RFD ≥ 10%, 80 met all criteria and are considered as founder variants either in the QcP, UQc or SLSJ, regardless of whether they were documented or not in the four

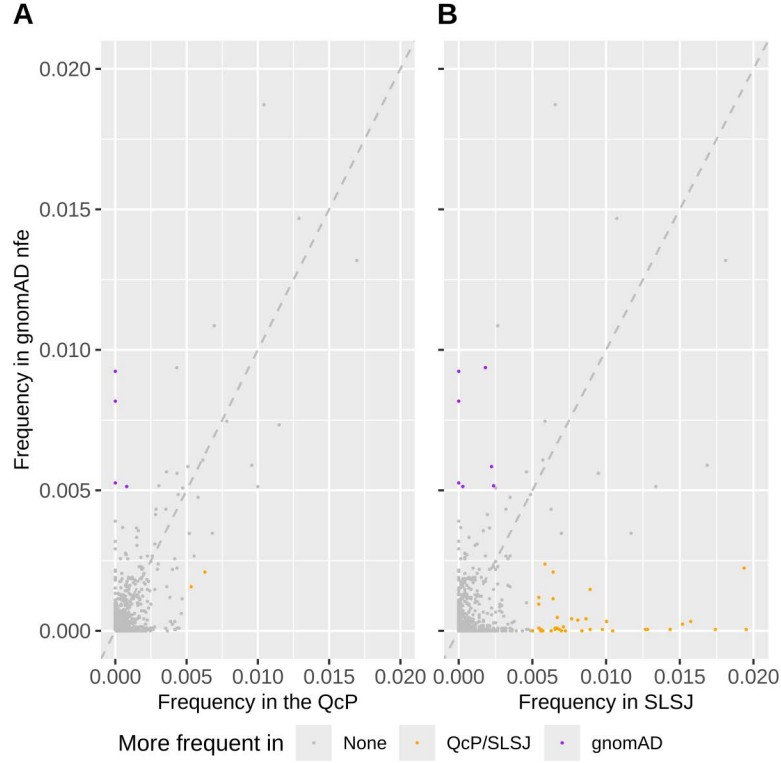

**Fig 1. Frequencies of 9,043 rare pathogenic variants in gnomAD nfe compared to A) QcP and B) SLSJ.** Only imputed variants also present in WGS were used.

aforementioned reviews [7–10]. Among these 80 founder variants, 38 were already documented in the four reviews [7–10] or in case reports (Table C in S1 Text), whereas 42 were never reported in the Quebec population (Tables 1 and S1).

### Founder variants' regional carrier rates and individuals' mutation load

We then compared the carrier rates of founder variants between the SLSJ and UQc (Fig 3). Most of the already reported founder variants are at higher CR than the newly identified ones, but some of the latter are as high as 1/22 in the SLSJ (Table 1). Carrier rates are generally higher in the SLSJ compared to the UQc. Specifically, the count of founder variants with carrier rates greater than 1/200 is eight times higher in SLSJ than in the UQc (three times higher when considering all variants with an RFD ≥ 10% regardless of whether they are founder variants) despite the lower sample size in SLSJ (Fig 4). Only 16 variants were at a higher CR in UQc than in the SLSJ among the 1,302 variants with RFD ≥ 10% despite the much greater sample size in UQc. Consequently, the number of individuals who carry at least one pathogenic founder variant is higher in the SLSJ than in the UQc ($chi^2$ p < 0.001) (Fig 5). In fact, for the variants already reported in the literature, 50% of the SLSJ and only 11% of the UQc individuals carry at least one variant. Notably, when the newly identified variants are added, these percentages reach 66% and 18%, respectively ($chi^2$ p < 0.001).

### SLSJ patients carrying newly identified founder variants

To confirm that our population-based method detects variants associated with diseases that are found in the SLSJ population, we requested clinical experts to examine their databases seeking for founder variants with CR above 1/200 (Table 1) that segregate within families of patients presenting the corresponding phenotype. Table 2 presents variants

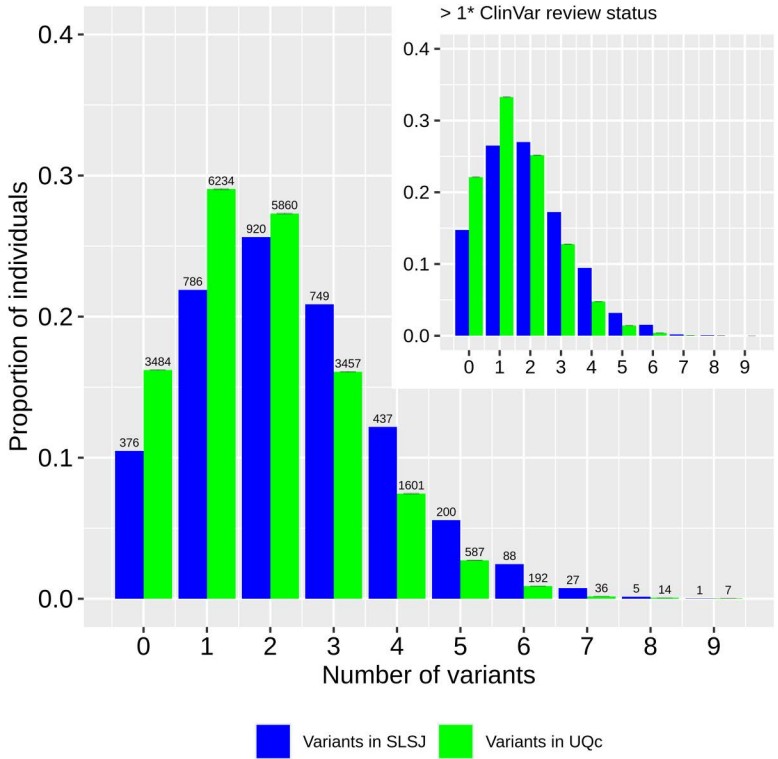

**Fig 2. Proportion of individuals carrying rare pathogenic variants.** The bars represent the average proportions obtained from 1,000 resamplings of 3,589 individuals from UQc, adjusted to match the sample size of the SLSJ. Error bars indicate the 95% confidence intervals. The numbers displayed above each bar correspond to the actual observed counts across all individuals. Only imputed variants also present in WGS were used. The inset shows the same analysis for variants with a ClinVar review status greater than 1 gold star.

found through different clinical panels in diagnosed patients from the Medical Genetics service (around 15,000 patients screened) and the *Clinique des maladies neuromusculaires* (CMNM; 45 medical records screened) of the CIUSSS of the SLSJ. We also had access to auto-reported CARTaGENE phenotypes (partial phenotyping for 30,000 participants). Of note, three of the variants identified in Table 2 (UROS c.217T>C, ETFA c.495_496del and CC2D2A c.4667A>T) in addition to two other variants not found at a homozygous state in the clinics (CEP290 c.7220_7223del and SGO1 c.67A>G) would be good candidates to include in an ongoing effort for designing a new carrier test for the SLSJ population in the Medical Genetics service [11]. A variant becomes of interest to the Medical Genetics service when it causes a severe disease without a curative treatment, and it would be considered appropriate to offer prenatal diagnosis with the possibility of medical termination of pregnancy. The variant must also be more frequent in the region, which is where our population-based approach proves valuable. It offers insight into the frequency and origin of the variant, even in the absence of clinical data.

## Discussion

In this study, we aimed to identify pathogenic variants found at higher frequency in the QcP and, more specifically, in the SLSJ region. We found 1,302 rare variants with RFD ≥ 10% in Quebec compared to gnomAD nfe. Among these, we identified 80 that met all criteria to be classified as founders, 38 of these being previously reported in the QcP within reviews or in case reports. Our study shows that establishing a systematic review of founder variants in a population is hard to conduct using only a literature review approach. This approach was done multiple times in the SLSJ population,

**Table 1. Novel founder variants found in this study.**

| Inheritance | Gene | Nucleotide | Disease name (ClinVar ID) | Data type | QcP (Sample sizes Imputed: 25,061 WGS: 1,852) | | UQc (Sample sizes Imputed: 21,472 WGS: 1,538) | | SLSJ (Sample sizes Imputed: 3,589 WGS: 314) | |
|---|---|---|---|---|---|---|---|---|---|---|
| | | | | | Count | CR | Count | CR | Count | CR |
| AD/AR | DNAH8 | c.8635_8636del | Primary ciliary dyskinesia (2037549) | Imputed data | 224 | 1/115 | 55 | 1/390 | 169 | 1/22 |
| AR | CNGA1 | c.947C>T | Retinitis pigmentosa 49 (16932) | Imputed data | 217 | 1/119 | 78 | 1/275 | 139 | 1/27 |
| AR | CTU2 | c.881C>A | Dysmorphic facies, renal agenesis, ambiguous genitalia, microcephaly, poly-dactyly, and lissencephaly (2067774) | Imputed data | 194 | 1/129 | 81 | 1/265 | 113 | 1/32 |
| AR | TMEM107 | c.*759C>T | Leukoencephalopathy with calcifications and cysts (265788) | Imputed data | 195 | 1/129 | 125 | 1/172 | 70 | 1/51 |
| AR | ENPP1 | c.583T>C | ENPP1-related disorder (2580630) | WGS | 7 | 1/309 | 1 | 1/1,538 | 6 | 1/52 |
| AD/AR | RGS9 | c.895T>C | Leber congenital amaurosis (5862) | Imputed data | 112 | 1/224 | 54 | 1/398 | 58 | 1/62 |
| AR | TRIOBP | c.1933C>T | Autosomal AR nonsyndromic hearing loss 28 (620162) | Imputed data | 112 | 1/224 | 64 | 1/336 | 48 | 1/75 |
| AR | UROS | c.217T>C | Cutaneous porphyria (3750) | Imputed data | 84 | 1/298 | 36 | 1/596 | 48 | 1/75 |
| AR | ASPM | c.8191_8192del | Microcephaly 5, primary, autosomal AR (21613) | Imputed data | 79 | 1/317 | 32 | 1/671 | 47 | 1/76 |
| AR | PYGM | c.148C>T | Glycogen storage disease, type V (2298) | Imputed data | 151 | 1/166 | 109 | 1/197 | 42 | 1/85 |
| AR | CEP290 | c.7220_7223del | Meckel syndrome, type 4|Bardet-Biedl syndrome 14 (418123) | Imputed data | 70 | 1/358 | 30 | 1/716 | 40 | 1/90 |
| AR | DONSON | c.1047-9A>G | Microcephaly, short stature, and limb abnormalities (431414) | Imputed data | 51 | 1/491 | 12 | 1/1,789 | 39 | 1/92 |
| AD | PKD1 | c.9829C>T | Polycystic kidney disease, adult type (192320) | Imputed data | 86 | 1/291 | 47 | 1/457 | 39 | 1/92 |
| AR | ETFA | c.495_496del | Multiple acyl-CoA dehydrogenase defi-ciency (459956) | Imputed data | 96 | 1/261 | 61 | 1/352 | 35 | 1/103 |
| AR | DNAH9 | c.1733del | DNAH9-related disorder (3013954) | Imputed data | 54 | 1/464 | 20 | 1/1,074 | 34 | 1/106 |
| AR | MOCOS | c.2326C>T | Xanthinuria type II (253162) | Imputed data | 57 | 1/456 | 24 | 1/895 | 33 | 1/116 |
| AR | CCDC40 | c.961C>T | Primary ciliary dyskinesia 15 (216118) | Imputed data | 35 | 1/716 | 7 | 1/3,067 | 28 | 1/128 |
| AR | SLC26A4 | c.1001+1G>A | Pendred syndrome (4819) | Imputed data | 52 | 1/482 | 25 | 1/859 | 27 | 1/133 |
| AD/AR | EIF2AK4 | c.1153dup | Familial pulmonary capillary hemangio-matosis (101527) | Imputed data | 34 | 1/737 | 8 | 1/2,684 | 26 | 1/138 |
| AR | TSHB | c.373del | Isolated thyroid-stimulating hormone deficiency (437070) | Imputed data | 40 | 1/627 | 14 | 1/1,534 | 26 | 1/138 |
| AR | DYNC2I2 | c.1312_1313del | Short-rib thoracic dysplasia 11 with or without polydactyly (665979) | Imputed data | 183 | 1/137 | 158 | 1/136 | 25 | 1/144 |
| AR | PHKB | c.1257T>A | Glycogen storage disease IXb (13620) | Imputed data | 30 | 1/835 | 6 | 1/3,579 | 24 | 1/150 |
| Unknown | CDK5RAP2 | c.2202+1G>A | not provided (1066422) | Imputed data | 100 | 1/251 | 77 | 1/279 | 23 | 1/156 |
| AD/AR | KCNJ1 | c.472G>A | Bartter syndrome (2506156) | Imputed data | 36 | 1/696 | 13 | 1/1,652 | 23 | 1/156 |
| AR | ASAH1 | c.410A>G | Spinal muscular atrophy-progressive myoclonic epilepsy syndrome (375548) | Imputed data | 95 | 1/267 | 73 | 1/298 | 22 | 1/163 |
| Unknown | DCAF6 | c.2240G>A | Cerebral visual impairment and intellec-tual disability (224814) | Imputed data | 83 | 1/302 | 61 | 1/352 | 22 | 1/163 |
| AR | PKHD1 | c.6793C>T | Autosomal AR polycystic kidney disease (1946278) | Imputed data | 129 | 1/194 | 107 | 1/201 | 22 | 1/163 |
| AR | TYR | c.572del | Tyrosinase-negative oculocutaneous albinism (99570) | Imputed data | 63 | 1/411 | 41 | 1/551 | 22 | 1/163 |
| AR | ALMS1 | c.11648_11649 insGTTA | Alstrom syndrome (550627) | Imputed data | 93 | 1/269 | 72 | 1/298 | 21 | 1/171 |

*(Continued)*

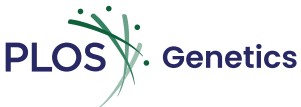

**Table 1.** (Continued)

| Inheritance | Gene | Nucleotide | Disease name (ClinVar ID) | Data type | QcP (Sample sizes Imputed: 25,061 WGS: 1,852) | | UQc (Sample sizes Imputed: 21,472 WGS: 1,538) | | SLSJ (Sample sizes Imputed: 3,589 WGS: 314) | |
|---|---|---|---|---|---|---|---|---|---|---|
| | | | | | Count | CR | Count | CR | Count | CR |
| AR | ERCC2 | c.2164C>T | Cerebrooculofacioskeletal syndrome 2 (16792) | Imputed data | 24 | 1/1,044 | 3 | 1/7,157 | 21 | 1/171 |
| Unknown | RAD50 | c.3779del | Hereditary cancer-predisposing syndrome (185537) | Imputed data | 27 | 1/928 | 6 | 1/3,579 | 21 | 1/171 |
| AR | SLC45A2 | c.264del | Oculocutaneous albinism type 4 (242518) | Imputed data | 77 | 1/325 | 56 | 1/383 | 21 | 1/171 |
| AR | RMRP | n.71A>G | Metaphyseal chondrodysplasia, McKusick type (14208) | Imputed data | 125 | 1/200 | 105 | 1/204 | 20 | 1/179 |
| AD | CHEK2 | c.247del | Hereditary cancer-predisposing syndrome (142851) | Imputed data | 29 | 1/864 | 10 | 1/2,147 | 19 | 1/189 |
| AR | NPHS1 | c.2071+2T>C | Finnish congenital nephrotic syndrome (56460) | Imputed data | 38 | 1/660 | 19 | 1/1,130 | 19 | 1/189 |
| Unknown | PKLR | c.1091G>A | PKLR-related disorder (1456959) | Imputed data | 97 | 1/258 | 78 | 1/275 | 19 | 1/189 |
| AR | ACY1 | c.575dup | Aminoacylase 1 deficiency (800812) | Imputed data | 58 | 1/448 | 40 | 1/565 | 18 | 1/199 |
| AD/AR | CAPN3 | c.2115+1G>A | Autosomal AR limb-girdle muscular dystrophy type 2A (555599) | Imputed data | 26 | 1/964 | 8 | 1/2,684 | 18 | 1/199 |
| AR | GMPPB | c.79G>C | Autosomal AR limb-girdle muscular dystrophy type 2T (60546) | Imputed data | 47 | 1/533 | 29 | 1/740 | 18 | 1/199 |
| AR | NDUFV1 | c.1162+4A>C | Mitochondrial complex I deficiency, nuclear type 1 (372716) | Imputed data | 52 | 1/482 | 34 | 1/632 | 18 | 1/199 |
| AR | RSPH3 | c.859+1G>T | Primary ciliary dyskinesia 32 (2980542) | Imputed data | 35 | 1/716 | 17 | 1/1,263 | 18 | 1/199 |
| Unknown | LARS1 | c.2500A>T* | not specified (3117894) | Imputed data | 124 | 1/202 | 109 | 1/197 | 15 | 1/239 |

AD: Autosomal dominant, AR: Autosomal recessive, SLSJ: Saguenay–Lac-Saint-Jean, UQc: Urban Quebec regions, QcP: Quebec Province, WGS: Whole-genome sequencing, CR: Carrier rate. ∗ Founder variant only in UQc.

and our results show that several founder variants with high carrier rates were missed. Moreover, there is currently no universally accepted definition of a founder variant in the literature. Commonly, a variant is considered to be associated with a founder effect when it is observed at elevated frequency within a genetically related group and can be traced to one or more common ancestors [4,12,13]. In our study, in addition to the frequency criteria, we assessed IBD sharing at the variant locus among carriers, which indicates that it was inherited from a recent common ancestor.

In addition to confirming known founder variants, we also report for the first time 42 novel founder variants that, to our knowledge, have never been documented in the QcP. Some of these exhibit a high carrier rate, comparable to the six diseases included in the carrier test offered to the population. These new variants could potentially account for unreported rises in disease prevalence within the population, which suggests a potential underestimation of the overall prevalence of rare diseases in the SLSJ region, as also reported in other populations with founder effects [14,15]. Indeed, adding the newly identified founder variants raises the proportion of individuals carrying at least one pathogenic founder variant by 1.3 and 1.7 times in the SLSJ and in the UQc, respectively. These proportions are similar when considering only ClinVar variants with stronger evidence of pathogenicity.

Recent results from the carrier testing currently offered to the population revealed that 116 couples identified as carriers of the same variant were provided with genetic counselling to discuss their reproductive options [11]. Given that the incidence of the four conditions currently included in the carrier test is approximately 1/2,000 births [16–19] and that around



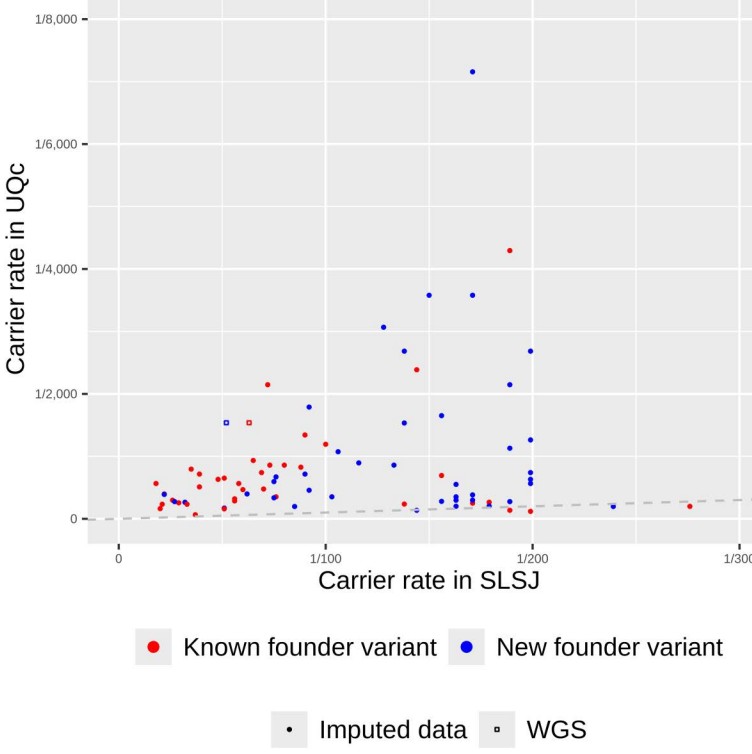

**Fig 3. Carrier rates for founder variants.** Only variants classified as founders in SLSJ, UQc or QcP are shown here (80 variants).

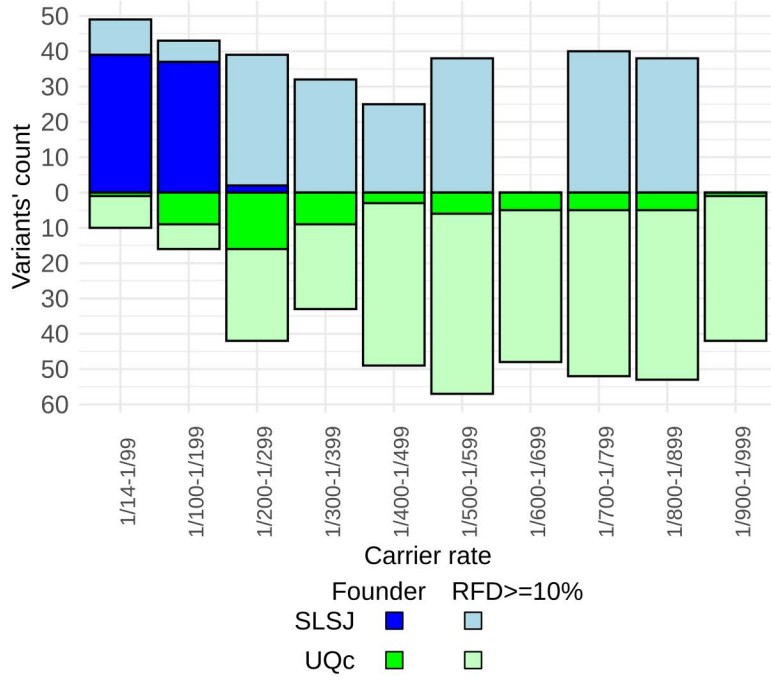

**Fig 4. Number of variants according to their carrier rate.** Only the CR from the imputed data was used. Note that the 78 founder variants shown in dark colors may originate from either group, which explains why some exhibit a CR below 1/200, indicating they are founder exclusively in the other group.

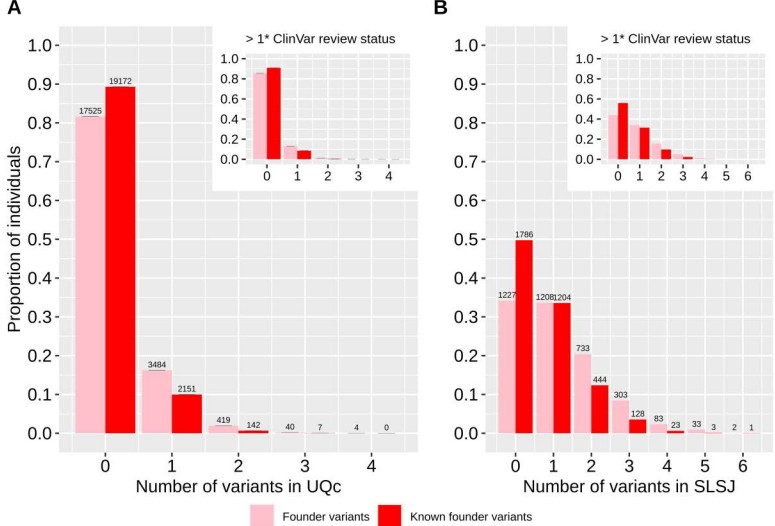

**Fig 5. Proportion of individuals carrying founder variants in A) UQc and B) SLSJ.** Only rare variants classified as founders in SLSJ or UQc or QcP in imputed data are shown here (78 variants). The bars represent the average proportions obtained from 1,000 resamplings of 3,589 individuals from UQc (A), adjusted to match the sample size of the SLSJ. Error bars indicate the 95% confidence intervals. The numbers displayed above each bar correspond to the actual observed counts across all individuals. The inset shows the same analysis for variants with a ClinVar review status greater than 1 gold star.

**Table 2. Variants found in patients with corresponding phenotypes.**

| Inheri-tance | Gene | Nucleotide | Phenotype | Heterozygotes | Homozy-gotes | Clinic |
|---|---|---|---|---|---|---|
| AD | *PRPH2* | c.554T>C | Retinitis pigmentosa | 1 | 0 | Genetic |
| AR | *CC2D2A** | c.4667A>T | Joubert syndrome | – | 1 | Genetic+CMNM |
| AR | *PDZD7* | c.2107del | Hearing loss, autosomal recessive 57 | 4 (compound with c.2672AGA[1]) | 0 | Genetic |
| AR | *EIF2AK4* | c.1153dup | Familial pulmonary capillary hemangiomatosis | – | 1 | Genetic |
| AR | *SLC45A2* | c.264del | Oculocutaneous albinism type 4 | – | 1 | Genetic |
| AR | *TYR* | c.572del | Tyrosinase-negative oculocutaneous albinism | 2 (compound with c.1046G>C) | 1 | Genetic+CaG |
| AR | *ALMS1* | c.11648_11649 insGTTA | Alstrom syndrome | – | 1 | Genetic |
| AR | *ETFA** | c.495_496del | Multiple acyl-CoA dehydrogenase deficiency | – | 1 | Genetic |
| AR | *UROS** | c.217T>C | Cutaneous porphyria | 3 (compound with c.424C>T) | 0 | Genetic |
| AR | *SLC26A4* | c.1001+1G>A | Pendred syndrome | – | 3 | Genetic |
| AD | *CHEK2* | c.247del | Cancer | 10 | 0 | Genetic+CaG |
| AD | *PKD1* | c.9829C>T | Polycystic kidney disease, adult type | 1 | 0 | CaG |

Variants in gray were reported in Quebec, but not in SLSJ, while other variants were not reported, *: Considered for a new carrier test in the SLSJ, Heterozygotes: Number of heterozygous patients (for dominant diseases), Homozygotes: Number of homozygous patients (for recessive diseases), CaG: CARTaGENE phenotypes.

2,000 births occur annually in the SLSJ region, genetic testing has the potential to significantly reduce disease incidence. Expanding the carrier screening panel to include additional conditions could greatly enhance public health outcomes. Indeed, we found patients carrying newly described founder variants who have been diagnosed with the corresponding disease in clinical databases. Five of the variants found in this study would be good candidates for inclusion in a carrier

test, which would have the potential to detect more couples who are carriers of the same condition. Importantly, those five variants were not previously recognized as more prevalent in the region through clinical observation alone, highlighting the value of our population-based approach. Establishing carrier rates plays a critical role in advancing precision medicine and carrier testing among populations with a founder effect [14]. In addition, it is a great proof-of-concept for larger initiatives to come in the field of precision medicine in regard to carrier frequency panels in larger populations.

We demonstrate an underestimation of the number of pathogenic variant carriers in SLSJ, which has been the focus of numerous studies on rare genetic diseases linked to the founder effect and population expansion. This supports the hypothesis of a higher mutation load (which we define here as a rise in deleterious allele frequencies) following range expansions [20] and under certain demographic and dominance models [21]. Consequently, the number of individuals affected by a rare disease might be underestimated in many countries or local communities. Our population cohort's approach could be applied in other worldwide populations at low costs, thus helping in enhancing and accelerating the molecular diagnosis of patients.

The present study demonstrates the higher pathogenic mutational load in individuals from the SLSJ region compared to UQc, not only for founder variants, but also for all ClinVar variants found in the QcP. It seems that the SLSJ individuals are more likely to carry at least one variant, despite the greater loss of variants in the SLSJ compared to UQc. The higher mutation load in the SLSJ individuals is mainly caused by an overrepresentation of variants with a CR greater than 1/200. This is the result of the very rapid population expansion, five times greater than the one observed in the whole Quebec for the same period [22]. Some founders in SLSJ contributed a lot to the present population [23] and therefore could have introduced an allele in the population that would reach such a high frequency [24]. Moreover, it was demonstrated that the first SLSJ settlers had an increased fitness [25], which could have contributed to increasing deleterious allele frequencies [26,27].

This study has certain limitations. Firstly, the sample size of the WGS data may be insufficient to accurately estimate the frequency of variants in the population. Therefore, we chose to work with imputed data, which includes a significantly larger number of individuals. To achieve the most accurate representation of our data, especially given our focus on rare variants, we performed imputation using a local WGS rather than a global worldwide reference panel. However, we acknowledge that imputed data may not be as reliable as WGS or genotyping. Therefore, we selected the ClinVar pathogenic variants exclusively in WGS, then we compared the WGS with the imputed data for the same individuals and excluded any unreliable imputed variants. We also excluded any individual whose cluster in WGS did not match the one in the imputed data clustering based on the UMAP. Also, our definition of a founder variant is stringent, with a CR of at least 1/200, especially for the SLSJ region, where the sample size is smaller. As a result, some less common but genuine founder variants might be missed, making this study somewhat conservative in the identification of founder variants.

These findings might be crucial for clinicians to shorten the patients' diagnostic odyssey and reduce the economic burden associated with undiagnosed rare diseases. This could help improve the management of patients and, for some of them, enhance their quality of life as appropriate follow-up could be offered earlier. With this information, precision medicine can implement targeted genetic screening programs, allowing for early detection of inherited conditions that are more prevalent due to the particular genetic structure shaped by some demographic events, such as a founder effect or a population expansion. This enables tailored prevention strategies, personalized treatments, and risk-reduction measures that are specific to the genetics of the population. Ultimately, analyzing carrier rates in populations could help healthcare providers offer more precise and effective medical care, enhancing outcomes for both individuals and the community. Indeed, we identified 30 patients carrying 12 causal variants that have not been previously reported as more frequent in SLSJ. The underestimation of pathogenic mutational load might also happen in other populations as a result of range expansions and rare diseases might be much less rare than anticipated. In an era of precision medicine with at least 10% of the population affected by rare diseases, it is crucial to adopt new approaches to enhance and accelerate the molecular diagnosis of rare diseases.

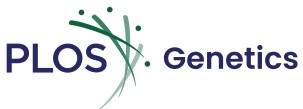

## Subjects and methods

### Ethics statement

This population-based study on the CARTaGENE cohort was approved by the University of Quebec in Chicoutimi (UQAC) ethics board. The approval for the secondary use of anonymized samples coming from the provincial screening testing was obtained from the Centre intégré universitaire de santé et de services sociaux du Saguenay—Lac-Saint-Jean of the SLSJ Direction of professional services. Written informed consent for the use of saliva samples for genetic testing was obtained from participants. The search of the clinical database for patients carrying the newly identified variants was approved by the Centre intégré universitaire de santé et de services sociaux du Saguenay—Lac-Saint-Jean ethics board.

### Cohort

The CARTaGENE cohort [28] (https://cartagene.qc.ca/) used in this study includes WGS of 2,184 and genotyping of 29,353 participants. Individuals aged between 40 and 69, residing in six distinct cities (Montreal, Quebec City, Trois-Rivières, Sherbrooke, Gatineau, Saguenay), were recruited between 2009 and 2015, regardless of their birthplace. The CARTaGENE cohort also includes a wide range of phenotypes, among which is the occurrence of a disease. For further details on genotyping and WGS data, see https://cartagene.qc.ca/files/documents/other/Info_GeneticData3juillet2023.pdf. All genomic data were aligned on the GRCh38 genome assembly.

### Genotypes cleaning and imputation

To increase our sample size and achieve more accurate carrier rates, we imputed the six different CARTaGENE genotyping chips using SHAPEIT5 [29] and IMPUTE5 [30] with default settings. The individuals were genotyped on different arrays (Omni 2.5, GSAv1 + Multi disease panel, GSAv1, GSAv2 + Multi disease panel, GSAv3 + Multi disease panel, GSAv2 + Multi disease panel + addon and Affymetrix Axiom 2.0) and were cleaned and merged as follows. Each dataset was cleaned separately using PLINK software v1.9 [31], ensuring individuals with at least 95% genotypes among all SNPs were retained. At the SNP level, we retained SNPs with at least 95% genotypes among all individuals, located on the autosomes and in Hardy–Weinberg equilibrium $p > 10^{-6}$.

The imputation was performed using 2,390 WGS from CARTaGENE and in-house Quebec samples [32] as a reference to enhance our capacity to identify rare variants within our population (refer to Table D in S1 Text for a comparison of imputations using either the local Quebec or TOPMed reference panel). Both WGS cohorts were jointly called using illumina DRAGEN v4.0 with popex tool. Variants with at least 10% missing genotypes, monomorphic variants, and variants in centromeres and in the ENCODE blacklist were filtered out from the WGS before performing imputation on each genotyping batch separately. All imputed genotyping batches were then merged, and the final imputed dataset includes 29,353 individuals. A post-imputation quality control filter was applied on each individual imputed batch to remove variants with an imputation quality score <0.3 for the PCA and UMAP.

### UMAP and clustering according to individuals' origin

For the purpose of this study, we needed to identify clusters of individuals based on genetics, regardless of where they were recruited. To do so, a PCA was performed using PLINK on the WGS SNPs with a minor allele frequency (MAF) of at least 5% and after removal of SNPs with more than 2% missing individuals and in LD (--indep-pairwise 200 5 0.1). We retained only biallelic SNPs within the accessibility mask [33], resulting in a total of 90,073 remaining SNPs. We also filtered out individuals with more than 2% missing SNPs, resulting in 2,166 individuals remaining. A UMAP [34] was then performed on the first three PCs (determined by the scree test) with the R umap library v0.9.2.0 (Fig B in S1 Text). This technique was proven efficient to reveal fine-scale population structure [35]. K-means clustering was then employed to create three clusters, aiming to retain as many individuals from the SLSJ as possible, given its limited sample size. We

also intended to choose individuals with the strongest ancestry connection to the region. Based on the recruitment place (Fig B, panel A in S1 Text), we could see that the majority of the CARTaGENE participants recruited from the SLSJ region belong to the red cluster (Fig B, panel B in S1 Text). We identified 314 individuals originating from the SLSJ region (red cluster) and 1,538 individuals from the other urban Quebec regions (UQc) (green cluster), for a total of 1,852 for the QcP (green and red clusters). Clusters were also defined on imputed data as described above on pruned SNPs (--indep-pairwise 50 5 0.2) at 5% frequency or more, keeping five PCs for the UMAP, leaving 3,589 individuals in the SLSJ (red cluster) and 21,472 in the UQc (green cluster), for a total of 25,061 in the QcP (Fig C in S1 Text). This includes the 1,852 individuals with WGS that were also imputed from genotypes. The SLSJ cluster finally includes 90% of the individuals recruited from the SLSJ region for the WGS data and 84% for the imputed data. We ensured consistency of individuals in clusters between the WGS and imputed data by removing 27 samples that exhibited mismatches, likely because they were at the boundaries of both clusters. This method ensures that individuals have a common genetic background and has been shown to be helpful in uncovering rare variants with smaller sample sizes [36,37].

### Resampling of UQc

To minimize bias when comparing individual and variant numbers across both clusters, we performed 1,000 resamplings of 3,589 individuals from UQc. We then calculated the mean and 95% confidence interval (CI) for all 1,000 resamplings. The graphs display the average proportions or variants' number derived from these resamplings, with error bars indicating the 95% CI.

### Selection of pathogenic variants

Variants' classification was extracted from the ClinVar database version of June 24, 2024 [38]. Only variants classified as: Pathogenic, Likely pathogenic, and both pathogenic/likely pathogenic, as well as SNPs, insertions and deletions (indels), were included in the analysis, whereas repeat expansions were excluded. Furthermore, variants with the following review status were removed: no assertion criteria provided, no classification provided, and no classification for the individual variant. Additionally, we incorporated all variants referenced as founder variants in previous studies [7–10], regardless of their status on ClinVar. Therefore, we obtain a list of 240,716 variants.

9,043 pathogenic variants were present either in gnomAD or the QcP imputed data after removal of variants that were absent from the QcP WGS or unreliable in imputed variants (Table D in S1 Text). We also removed variants that were less frequent than 1/21,066 which is the lowest number of alleles (for gnomAD non_topmed_nfe). 1,590 imputed variants were present in the QcP.

### Calculation of relative frequency difference (RFD) ≥10%

We calculated the variants' frequency in the CARTaGENE WGS and imputed data using PLINK v1.9 for the individuals originating from the SLSJ, UQc and QcP (both SLSJ and UQc clusters) inferred by the clustering. The gnomAD frequencies for the non-Finnish Europeans (non_topmed_nfe) were directly extracted from gnomAD genomes v3.1.2. To calculate the RFD of a variant in the QcP compared to gnomAD nfe, we used the following formula:

$$RFD = \frac{freq_{QcP} - freq_{gnomAD}}{freq_{QcP}}$$

Knowing that:

• $freq_{QcP}$ corresponds to the frequency of the variant in the *QcP* population.

• $freq_{gnomAD}$ corresponds to the frequency of the variant in the gnomAD non-Finnish Europeans.

We fixed a minimum RFD threshold of 0.1 to make sure it encompasses a large number of variants that could be of interest, although at very low frequencies, the difference between both populations may be minimal and not distinguishable from sampling noise. For instance, RFD = 0.1 corresponds to a 1.11-fold increase and RFD = 0.5 to a 2-fold increase. When a variant with RFD ≥ 10% identified in the WGS was also found in the imputed data, we used the imputed variant frequency. If not, we relied on the WGS variant frequency, ensuring that the RFD was at least 10% in both data types. Notably, the frequencies of variants show a strong correlation between both data types (Fig D in S1 Text). We detected 1,304 potentially pathogenic variants that reached RFD ≥ 10% compared to gnomAD nfe. Since we are focusing on rare variants, we removed two variants with a MAF ≥ 5% (chr6:26090951:C:G and chr14:94380925:T:A), leaving 1,302 rare variants with RFD ≥ 10% in the QcP.

### Estimation of carrier rate (CR)

We directly counted the number of heterozygotes for each variant and determined the CR by calculating the frequency of the heterozygous individuals expressed as 1/x.

### IBD sharing at variants' location

All cleaned genotyping batches (excluding the Affymetrix chip due to its poor SNPs intersection with other Illumina chips) were combined and only the intersecting common SNPs were kept. After the merge, individuals with less than 95% genotypes among all SNPs and SNPs with less than 95% genotypes across all individuals were once again filtered out. The final dataset comprises 148,200 SNPs and 28,358 individuals.

We then inferred IBD segments on phased genotypes using refinedIBD [39] version 17Jan20 and Beagle version 18May20. Subsequently, the segments were merged using the merge-ibd-segments 17Jan20.102 tool. We retained only IBD segments of 2Mb or longer and with a LOD score greater than 3.

We then examined the proportion of pairs of individuals sharing IBD along the genome among carriers of a specific pathogenic variant in the SLSJ and UQc clusters separately. This proportion is generally around 0.02 in SLSJ individuals who are not closely related [40].

### Selection of founder variants

After selecting ClinVar pathogenic variants with RFD ≥ 10%, we established additional criteria for a variant to be considered as founder. The number of individuals carrying the variant must be adequate to avoid false signals or misinterpretations while also being high enough to be relevant for inclusion in population screening tests [7–10]; thus, the target CR was set to 1/200. Hence, given the different sample sizes, the minimum threshold is five (1/63), eight (1/192) and ten (1/185) individuals carrying the variant for WGS of SLSJ, UQc and QcP respectively. We chose not to include variants found in fewer than five individuals to minimize the risk of false signals and because IBD sharing patterns were unclear with such low counts. However, for imputed data, we set the threshold to reach a CR of 1/200, which represents 18, 108 and 126 individuals for the SLSJ, UQc and QcP. For instance, a variant with a carrier rate of 1/200 for a recessive disease in a population of 286,768 (the SLSJ population in 2024) would be expected to result in approximately seven affected individuals. Furthermore, to be called as founder, a variant must show a proportion of pairs of carriers sharing IBD around the variant of at least 0.5, indicating that half of the carriers' pairs share IBD at the variant's location. Considering that a founder variant usually originates from a single ancestor in a population with a founder effect [24], and within a relatively recent time frame to increase in frequency due to drift, this variant will still be in LD with other surrounding variants and carriers will share not only the variant, but also the surrounding haplotype. Therefore, examining the IBD sharing around a variant is a dependable method to confirm its recent common origin due to the founder effect and population expansion.

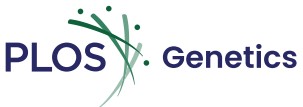

## Clinical data

The patient group consisted of individuals residing in SLSJ during the assessment, all of whom had genetic disorders. They were clinically evaluated at the Medical Genetics service and the CMNM of CIUSSS of SLSJ. Their DNA samples were analyzed in certified clinical molecular laboratories as part of the clinical testing and genetic evaluation process. A review of nearly 15,000 patients in internal databases and medical records enabled the identification of patients who were homozygous, compound heterozygous, or heterozygous for autosomal recessive or dominant variants.

## Supporting information

**S1 Table. Description of variants with RFD ≥ 10%.**
(XLSX)

**S1 Text.** Table A: Variants previously described in the population not found here. Table B: Experimental assessment of CR in an independent cohort of 1,000 individuals living in SLSJ. CI have been calculated on proportions using the online tool https://sample-size.net/confidence-interval-proportion/; * CR calculated using WGS in the present study; ** One sided 97.5% CI. Table C: Case reports on Quebec founder variants. Table D: Comparison of imputations of ClinVar rare variants using the TOPMed r2 or the Quebec reference panel. Proportions are based on common SNPs/genotypes except for missing SNPs which is on the total number of SNPs. False positives are defined as heterozygotes in imputed, but not in WGS data. False negatives are defined as heterozygotes in WGS, but not in imputed data. * In at least one individual. **SNPs having at least one homozygote switch genotype in addition to SNPs having more than 10% of false positive genotypes (101 out of the 106 false positive SNPs) were considered as unreliable in the imputed data. Fig A: Comparison of the variants' carrier rates reported in SLSJ and found in our analysis. When available, the aggregated CR (all variants associated with the same disease) was used; also, if available, the CR from the imputed data was used; otherwise, the CR from WGS data was utilized. Variants from the same disease were grouped as in previous studies. Fig B: UMAP of WGS data. UMAP are coloured according to A) the recruitment region or country of birth and B) the k-means clustering. Fig C: UMAP of imputed data. UMAP are coloured according to A) the recruitment region or continent of birth and B) the k-means clustering. Note that 1,537 pathogenic variants had an RFD ≥ 10% in the WGS data, but only 1,302 of them had also an RFD ≥ 10% in the imputed data. Fig D: Correlation between imputed and WGS variants' frequency in QcP.
(PDF)

## Author contributions

**Conceptualization:** Élisa Michel, Claudia Moreau, Laurence Gagnon, Simon L. Girard.

**Data curation:** Élisa Michel, Laurence Gagnon, Josianne Leblanc, Jessica Tardif, Mathieu Desmeules, Jean-Denis Brisson.

**Formal analysis:** Élisa Michel, Mylène Gagnon.

**Funding acquisition:** Cynthia Gagnon, Luigi Bouchard, Simon L. Girard.

**Project administration:** Simon L. Girard.

**Resources:** Lysanne Girard, Luigi Bouchard.

**Supervision:** Claudia Moreau, Simon L. Girard.

**Validation:** Lysanne Girard.

**Writing – original draft:** Élisa Michel, Claudia Moreau.

**Writing – review & editing:** Laurence Gagnon, Mylène Gagnon, Josianne Leblanc, Jessica Tardif, Lysanne Girard, Jean Mathieu, Cynthia Gagnon, Mathieu Desmeules, Jean-Denis Brisson, Luigi Bouchard, Simon L. Girard.

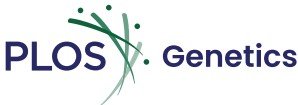

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
