## [Decision Letter · Decision Letter 0]

16 May 2025

PGENETICS-D-25-00284

Rare diseases load through the study of a regional population

PLOS Genetics

Dear Dr. Girard,

Thank you for submitting your manuscript to PLOS Genetics. After careful consideration, we feel that it has merit but does not fully meet PLOS Genetics's publication criteria as it currently stands. Therefore, we invite you to submit a revised version of the manuscript that addresses the points raised during the review process.

Please submit your revised manuscript within 60 days Jul 15 2025 11:59PM. If you will need more time than this to complete your revisions, please reply to this message or contact the journal office at plosgenetics@plos.org. Please include the following items when submitting your revised manuscript:

We look forward to receiving your revised manuscript.

Kind regards,

Jonathan Marchini

Academic Editor

PLOS Genetics

Gregory Cooper

Section Editor

PLOS Genetics

Aimée Dudley

Editor-in-Chief

PLOS Genetics

Anne Goriely

Editor-in-Chief

PLOS Genetics

**Journal Requirements:**

**Reviewers' comments:**

Reviewer's Responses to Questions

**Comments to the Authors:**

Reviewer #1: Remarks to the authors

Michel et al investigate the genetic burden of rare diseases in Quebec’s Saguenay–Lac-Saint-Jean (SLSJ) region, bridging previous population-genetics studies from the lab with recent studies on the disease burden in the region (Cruz Marino et al.). The authors perform within-cohort imputation to increase resolution, and, after some basic pop-gen analysis, identify two clusters - 3,589 individuals from SLSJ and 21,472 from UQc. Next, starting from 240,710 P/LP variants from ClinVar, they follow several cleaning steps and identify 80 pathogenic variants with founder effects in either SLSJ or UQc. 42 of these were not reported in previous studies of the Quebec population and, interestingly, a dozen of those were found to be carried by patients having the corresponding disease. The study is interesting and has much to contribute towards better disease diagnosis, though I believe there are a few aspects that could be improved.

Major comments

1. The analysis about clinical validation is very interesting, and perhaps the highlight of the paper, but it should be expanded. First, how large were the external databases, and is there any chance that patients found by clinicians were also used in the analysis (“train-test contamination”)? How many variants were sent for validation and how many of those ~30 people were previously undiagnosed? If any, that needs highlighting. Some of these questions might have obvious answers (or no answer due to privacy constraints), but more details should be given to help assess the diagnostic utility of the findings.

2. The way the authors compare their list of founder variants with those previously reported is confusing. They first describe 42/72 reported previously (and found here), then mention “38 of the (80) founder variants were already documented…”. How do the 38 variants compare with the “42 previously reported” given in Table 1? Please consider restructuring this section for clarity by first presenting your findings, then indicating which are novel and which overlap with prior studies.

3. A common choice for comparing two groups is a two-samples t-test. Instead, the authors report p-values from chi2 tests. This choice should be justified, and the statistical tests described in more detail in Methods.

4. Using IBD sharing to clean founder variants is an interesting feature, but how did that help on top of the CR filter? Is there a reference paper that did something similar? The authors should justify the use of 50% as threshold, and how sensitive their analysis is.

Minor comments

• The section “Experimental validation of carrier rates” is misleading, as experimental usually refers to lab work carried out to validate one of the main findings. Effectively what the authors did was targeted genotyping to verify that their statistical imputation was accurate. Nice to have, but I believe that is more of a supplementary analysis.

• Please convert Supplementary Table 1 to csv rather than PDF.

• L277: “most related individuals” is that a typo? PCA needs to be performed in a set of unrelated individuals, and then any relatives need to be projected on the principal components.

• Table captions should be expanded to include definitions (e.g., F/NF) and sample sizes, particularly for Table 1.

• There are references mentioned twice, e.g. 10 & 21 or 13 & 18, perhaps more.

• Consider citing Ishiki et al 2024 (PMID: 39399040) who performed a similar study.

• L319: should be 1,302 variants.

• P-values are not written in the correct format, or contain typos.

Reviewer #2: The authors have sought to provide a comprehensive account of potentially pathogenic variants which are more frequent in Quebec than in the gnomAD non-Finnish European (NFE) genomes, relying on both variants from ClinVar and previously identified founder variants from Quebec or specifically the Saguenay–Lac-Saint-Jean (SLSJ) region. Using carrier rate thresholds and an IBD-based analysis, the authors identified 80 potentially pathogenic founder variants in SLSJ or the rest of Quebec, some of which are novel and some of which were previously reported. A subset of potential founder variant carrier rates were validated using a TaqMan genotyping assay, and some variants were observed in medical databases in Quebec. Mutational load (based on carrier counts) was found to be greater in SLSJ than in the rest of Quebec, both of the potentially pathogenic founder variants, and of all potentially pathogenic variants more common in SLSJ/Quebec than in gnomAD NFE.

This manuscript should be commended for its database-first approach to detecting potentially pathogenic founder variants in SLSJ and the rest of Quebec, which importantly draws upon a comprehensive analysis of potentially pathogenic variants in ClinVar, as well as the existing literature on founder variants in SLSJ. The authors make valuable observations about the proportion of potentially pathogenic founder variants in SLSJ/Quebec which are previously reported/unreported in the literature, as well as the mutational load of these variants in the regions studied. The careful use of imputed genotypes and an IBD-based analysis of potential founder variants both strengthen the study’s key findings. Below I have listed several major and minor points of criticism, which deal primarily with manuscript clarity and additional analyses of the authors’ existing data.

Major Points:

1. Lines 99-101. Given that many more WGS and imputed samples were available for the UQc than SLSJ cluster, (314 WGS and 3,589 imputed from SLSJ, or 1,538 WGS and 21,472 imputed from UQc), it is unsurprising that many more of the RFD >= 10% variants would be unique to UQc vs SLSJ. Regarding this, the authors state in the Discussion that “Indeed, 42% of variants with an RFD≥10% in the QcP were lost in the SLSJ, although some of them might be too rare to be observed in the SLSJ due to the smaller sample size” (lines 200-201). To better compare the count or proportion of variants missing from SLSJ and UQc, the authors should as a separate analysis down-sample the UQc cohort to match the SLSJ sample size, or repeatedly bootstrap equal-size subsets.

2. Lines 99-101. Additionally, the cited missing variant counts (540 from SLSJ and 17 from UQc) are not shown in Figure 1, or anywhere else that I can find. It would be useful to provide a table summarising the counts of potentially pathogenic RFD >= 10% variants coming from SLSJ, UQc, and QcP, and how many are unique to that cluster.

3. Lines 99-101. It is possible to visually confirm from Figure 1 that many variants are much more frequent in SLSJ than gnomAD NFE, seen on the right half of Figure 1b close to the X axis, whereas this is not the case for the combined QcP cluster in Figure 1a. I am wondering, however, if there is a better way to quantify that “many variants are more frequent in the SLSJ region”, such as a table which counts the number or proportion of the 1,302 variants above a certain RFD or fold-increase threshold for each cluster? How many variants, for instance, have a >=50% or 90% RFD vs gnomAD NFE in the SLSJ or QcP clusters?

4. Lines 101-103. Figure 2 plots the count and proportion of individuals from the SLSJ and UQc (rest of Quebec) clusters which carry 0, 1, 2, 3 … 9 of the 1,302 rare potentially pathogenic >= 10% RFD variants. Given that this figure primarily seems to shed light on the mutational load in SLSJ vs UQc, I am wondering whether the >= 10% RFD threshold is necessary? Why not repeat the same analysis for all rare, potentially pathogenic variants (as defined by ClinVar and previous SLSJ publications), regardless of RFD threshold? Additionally, I find it striking that, for SLSJ, only ~13% of individuals carry none of the 1,302 rare, potentially pathogenic >= 10% RFD variants of interest (and ~27% of UQc individuals). Because many of these variants are likely to come from ClinVar P/LP entries with just 1* review status, it would be informative to repeat this analysis with only ClinVar P/LP variants with >= 2* review status (plus the previously reported founder variants).

5. Lines 113-115. I cannot find any mention elsewhere in the manuscript or table/figure legends as to how the nine variants were selected to have their carrier rates reassessed, and whether this is an adequate number of variants to use for validation. Additionally, the authors should clarify (either in lines 113-115 or lines 359-361) whether the “subset of 1,000 randomly selected samples with appropriate consent” are a subset of the CARTaGENE individuals from SLSJ used in the SLSJ WGS/imputed datasets, or other individuals from SLSJ not used previously in the study. The term “subset” from lines 113-115 quoted above suggests the former, but the title of Supplementary Table 3 “Experimental reassessment of CR in an independent cohort of 1,000 individuals living in SLSJ” suggests the latter. If the 1,000 SLSJ subset does in fact come from the CARTaGENE 314 WGS and 3,589 imputed genotypes from SLSJ, then the authors should report the precise % of matching genotypes between the validation subset and the original data.

Minor Points:

6. Lines 59-66. The first paragraph of the Introduction gives some background on the prevalence of rare diseases (thought to affect 10% of population, more than 10,000 rare diseases in Orphanet). It would be useful to specify that the authors are mainly concerned with rare Mendelian diseases.

7. Lines 133-135. The authors should comment on whether any variants were more common in UQc than in SLSJ, and how many.

8. Lines 136-139. Because inheritance patterns are reported for many of these 80 founder variants (some are shown in Table 2), it would be useful to estimate the expected count or rate of affected individuals, in addition to the observed carrier counts shown in Figure 5.

9. Figure 4. The X axis labels are unusual, each ending in an open square bracket where it should be a closed square bracket. Additionally, I am confused why many UQc founder variants are shown with carrier rates below 1/200, given that a carrier rate above 1/200 was a requirement for founder variants. The authors should specify whether the label “UQc founder variants” here means variants which are founder variants in UQc, or something else, like founder variants in any region with their UQc carrier rate.

10. Lines 154-156. The nucleotide change and phenotype are listed, but authors should also list the associated gene name.

11. Table 3. The table is described as having corresponding phenotypes, but no actual phenotypes are listed.

12. Table 3. Nine instances of compound heterozygotes in patients are documented, also referred to in lines 171-174 of the Discussion. It would be useful to note the other variant in each case, and also to comment on whether it was possible to determine whether the two observed variants were in trans (a true compound heterozygote) or in cis on the same haplotype.

13. Lines 272-273. The final imputed dataset appears to have more individuals (29,353) than the number of initial CARTaGENE chip genotypes (29,337, mentioned lines 252-253). I am wondering if this refers to the number of imputed genotypes + the WGS, after filtering for sample missingness—the language is ambiguous. However, the subsequent application of the imputed dataset to refine the Quebec allele frequencies suggests the imputed and WGS datasets were not merged, as the authors mention that imputed frequencies only superseded WGS frequencies when the variant was present in the imputed dataset.

14. Lines 299-302. I presume that “conflicting (both pathogenic and likely pathogenic variants)” simply refers to variants with an aggregate germline classification of “Pathogenic/Likely pathogenic”, and not to “Conflicting classifications of pathogenicity”, as the latter term implies variants with both pathogenic and uncertain/benign submissions. If so, I would advise removing the word “conflicting” from the authors’ description.

15. Methods for calculating RFD, lines 306-321. As far as I can tell, closely related individuals were not removed from the SLSJ, UQc and QcP cohort clusters used to calculate various Quebec variant frequencies. If this was done, it should be stated, and if not, the choice should be explained.

16. Methods for calculating RFD, lines 306-321. It would be useful to comment on the statistical significance of an observed RFD of 0.1 between any of the SLSJ, UQc, and QcP cohorts and the gnomAD v.3.1.2 NFE genomes, for a variant at some of the gnomAD or Quebec/SLSJ allele frequencies relevant to this study. Additionally, for ease of reading, the authors should briefly state in the Methods or Results what an RFD of 0.1 would translate to in terms of fold-difference in frequency between the SLSJ/Quebec population and gnomAD NFE, perhaps for other reference RFDs as well, such as RFD = 0.5 or RFD = 0.9. For instance, RFD = 0.1 corresponds to a 1.11x greater frequency and RFD=0.5 corresponds to a 2x greater frequency.

17. Line 320. It would be useful to name the two variants with MAF >= 0.05, and to state how many of the remaining variants have, for instance, MAF >= 0.01 or MAF >= 0.005.

18. Lines 328-331. The authors explain that alternative observed carrier count thresholds of 5, 8, and 10 were used for the SLSJ, UQc, and QcP WGS clusters, due to the smaller sample sizes. The authors should clarify how exactly thresholds of 5, 8, and 10 were chosen.

19. Methods for IBD analysis, lines 336-350. I believe the comparison of pairwise IBD sharing at variant sites was done for each cluster (SLSJ, UQc, and QcP), as founder variants are reported for each in the Results section. Reading this section, particularly lines 343-346, I initially had the impression that the analysis was only done for SLSJ. This should be clarified.

20. Supplementary Table 1. Although this table is not meant to be read in its entirety, it is basically useless in the format it was sent to me for finding out anything about a variant’s status in SLSJ, as the variant info (ClinVar ID, Position GRCh38, Data type) for each variant are on different pages (1-25) than the SLSJ columns (26-49). It would also be good to include the gene name or symbol for each variant.

Reviewer #3: Review of the manuscript: Rare diseases load through the study of a regional population

Overview:

The authors present a survey of pathogenic rare variants in Quebec, particularly the Saguenay–Lac-Saint-Jean region (SLSJ), based on whole-genome sequencing of ~2200 individuals and genotype imputation of 29k others. The authors specifically look for variants with evidence of pathogenicity, high prevalence, recent origin, and higher frequency in Quebec compared to other Europeans. They validated 38 previously reported variants and identifies 42 previously unreported variants.

Strengths:

The study is based on a very large dataset. The population is important to study because of the founder event and, consequently, the high prevalence of pathogenic variants. Thus, the study has important clinical implications for the management of genetic testing in this population, particularly preconception carrier screening. The methods used to detect the founder variants are overall sound. The IBD analysis is particularly innovative and important for the identification of variants of recent origin.

Major comments:

• The results have implications for preconception carrier screening in Quebec and particularly in SLSJ. However, the authors don’t make specific recommendations for how such a panel should look like, which, unfortunately, somewhat diminishes the impact of the paper. It would be important to describe how such a panel would compare to existing panels in terms of the variants included and the number of couples at risk. (For example, see, https://www.nature.com/articles/gim2015123, although it is a bit dated and using a much smaller dataset).

• Some of the analyses are difficult to interpret, given that they are based on the same dataset used for discovery. The variants discovered are conditioned, by definition, to have certain frequencies in the discovery dataset. Thus, all results reporting variant counts (Figures 2-5) are somewhat biased (regression to the mean/winner’s curse). In other words, variants that made it to the list may have happened to be those “lucky” enough to cross the frequency threshold. However, in another dataset, their frequency will be lower. This is easily solvable by dividing the dataset into discovery and analysis, whereby variants will be discovered in the (larger) discovery subset, and variant counts will be reported based on the (smaller) analysis subset. I would also suggest that the analysis subset will have an identical number of genomes from SLSJ and UQc. Otherwise, again, all variant counts are difficult to interpret.

• It is difficult to justify the criterion of having a relative frequency difference greater than 10%. Figure 1 shows clearly that many variants are quite common in other Europeans, but have simply drifted to somewhat higher frequencies in Quebec. I would not call these founder variants. The founder variants are those on or close to the x-axis in Figure 1, namely variants with near-zero frequency outside Quebec that have reached a considerable frequency in Quebec. I think that more reasonable criteria would be 5x or 10x fold higher frequency in Quebec, along with frequency in gnomAD under (say) 1/1,000 or 1/10,000. Or the authors could use a method such as this paper: https://journals.plos.org/plosgenetics/article?id=10.1371/journal.pgen.1007329. Perhaps more generally, defining a first set of variants with RFD>0.1 and then a second set that also has a minimal carrier rate and high levels of IBD is quite confusing. I think it is really the second set that is interesting. The IBD analysis is very nice and important in this context.

• The clinical data analysis is missing many details. What was the sample size? Was there ethical approval for the search? Who did the search? Which variants were considered? In addition, Table 3 seems to contradict the associated text, as all the three variants mentioned in the text were seen only once based on Table 3, while line 159 suggests a variant is of medical interest only if seen three times. In Table 3, which variants are already in screening panels? More broadly, it is not clear what is the motivation for this analysis and what we actually learn from it. These variants are known to be pathogenic and they were found in the discovery dataset. So what new information do we learn from this review of medical records?

Minor comments:

• Lines 99-100: this result is likely due to the imbalance in sample size between SLSJ and UQc.

• Please add the y=x line to Figures 1 and 3.

• Table 1. What is the “status” column? Please also add the genomic coordinates of the variants (at least in a supplementary table, if space in the main text is limited).

• It will be interesting to see the distribution of IBD fraction across the 1302 variants of the first set and the 80 variants of the second set. This will help understand whether and how this filter is helping to identify the founder variants.

• Lines 155-156: please specify the meaning of the numbers in parenthesis.

• Please add to Table 3 the disease associated with each variant.

• Lines 171-176: we expect that by pure luck, some previously discovered variants will not be discovered in the current study above a certain frequency. The probability for this can be computed using binomial distributions.

• Lines 182-186: similar work has been performed in other populations. See, for example, https://www.nature.com/articles/s41436-019-0676-x.

• Lines 190-191: there is a very long literature on the question of whether founder populations have an increased load of deleterious variants. It is not obviously true, because, while founder events increase the frequency of some variants, they eliminate all the variants in people not surviving the founder event. In particular, see https://www.nature.com/articles/ng.2896 and the reviews https://www.sciencedirect.com/science/article/abs/pii/S0959437X14001002 and https://www.nature.com/articles/nrg3931. There were also later papers.

• Lines 193/245: I think that “fasten” and “fastening” are not used correctly.

• Supplementary figure 2: it is difficult to see how many variants are at each frequency given that points are overlapping.

• Lines 222-223: maybe I missed it, but I didn’t see anywhere mentioned filtering by 10% false positives.

• Line 267: could you please provide details on the in-house Quebec reference panel?

• Line 279: what were the parameters for removal of variants in LD?

• Line 294: how many samples were both the WGS dataset and in the imputed dataset? How many were only in the chip and are under a risk of misclassification?

• Line 295: the possibility of a mix-up is unclear to me – assuming these are samples with both WGS and chip data, why not just compare the data between the platforms and verify it’s the same person?

• Line 325: the carrier rate is just f_(hetero). Not the inverse.

• Line 358: which variants were validated using the TaqMan assay?

**Have all data underlying the figures and results presented in the manuscript been provided?**

Reviewer #1: **No: ** Sup Table 1 should be uploaded as csv/txt format and definitely not as PDF.

Reviewer #2: Yes

Reviewer #3: Yes

PLOS authors have the option to publish the peer review history of their article (what does this mean? ). If published, this will include your full peer review and any attached files.

**Do you want your identity to be public for this peer review?** For information about this choice, including consent withdrawal, please see our Privacy Policy .

Reviewer #1: No

Reviewer #2: No

Reviewer #3: No

**Figure resubmission:**
---

## [Decision Letter · Decision Letter 1]

6 Aug 2025

PGENETICS-D-25-00284R1

Rare diseases load through the study of a regional population

PLOS Genetics

Dear Dr. Girard,

Thank you for submitting your manuscript to PLOS Genetics. After careful consideration, we feel that it has merit but does not fully meet PLOS Genetics's publication criteria as it currently stands. Therefore, we invite you to submit a revised version of the manuscript that addresses the points raised during the review process.

Please submit your revised manuscript within 30 days Sep 05 2025 11:59PM. If you will need more time than this to complete your revisions, please reply to this message or contact the journal office at plosgenetics@plos.org. Please include the following items when submitting your revised manuscript:

We look forward to receiving your revised manuscript.

Kind regards,

Jonathan Marchini

Academic Editor

PLOS Genetics

Gregory Cooper

Section Editor

PLOS Genetics

Aimée Dudley

Editor-in-Chief

PLOS Genetics

Anne Goriely

Editor-in-Chief

PLOS Genetics

**Reviewers' comments:**

Reviewer's Responses to Questions

**Comments to the Authors:**

Reviewer #1: I am pleased that the authors have addressed all of my previous comments comprehensively, and the manuscript has been significantly improved as a result.

I have only two minor suggestions that would further enhance the manuscript's clarity, though these are very minor and so do not warrant another round of review:

1. The sentence "Based on the absence of a clear definition for founder variants, we propose here a new definition of a founder variant" would benefit from briefly mentioning what that definition is within the same sentence or immediately following it.

2. The section "Experimental validation of carrier rates" appears to interrupt the flow between figures and might be better positioned in the Supplementary Information to improve the manuscript's organization and readability.

Otherwise, I am happy to recommend this manuscript for publication in PLOS Genetics and believe it offers an important contribution to the field.

Reviewer #2: Brief Summary

I am grateful to the authors for their work responding to the major and minor points I have raised. The revision addresses most prior concerns: the authors have addressed sample-size imbalance by downsampling UQc, improved presentation of data, and clarified parts of their methodology. My outstanding requests described below relate primarily to the authors’ mutation load analyses: their use of the RFD>=10% filter for Figures 1 & 2, and their choice not to add a sensitivity analysis restricted to ClinVar >=2* variants for Figures 2 & 5. For these requests, see major point #4 and minor point #16 below. Additionally, I have suggested several minor text edits and clarifications—see major points #3 and #5, and minor point #6. If these items are addressed, I would consider my comments resolved.

Major Points:

1. Reviewer assessment: Adequately addressed. For Figures 2 & 5, the authors have adequately accounted for the difference in size between SLSJ and UQc by downsampling 3,589 imputed UQc individuals (the size of the imputed SLSJ dataset) 1,000 times.

2. Reviewer assessment: Adequately addressed. The authors have provided Supplementary Table 1 in a more readable format.

3. Reviewer assessment: Partially addressed. The authors are correct to observe that differences in mutation load are not primarily explained by differences in RFD or fold-difference, but rather differences in variant frequency. I am satisfied that the differences in variant frequency (or CR) are adequately described later in the manuscript, primarily in Figure 4. However, it would still be good to somehow quantify the observation on lines 104-105, which cites Figure 1. For instance, something like, “of the variants shown in Figure 1, __% had a frequency above X in QcP, whereas __% had a frequency above X in SLSJ.”

4. Reviewer assessment: Not addressed (rationale provided).

10% RFD: The authors have explained that the 10% RFD threshold was used to narrow down the ~200,000 ClinVar P/LP variants, and that including the variants with RFD < 10% would not change their observations about mutation load substantially, because most of them were rare, and/or not present in the imputed data. The “~200,000 ClinVar pathogenic variants” is misleading, however; the current version of Supplementary Figure 2 shows there are ~2,000 total ClinVar P/LP variants present in both QcP and gnomAD—I assume there are more found in just QcP but not gnomAD. It would be more accurate to say that the RFD threshold was used to narrow down to 1,537 (later 1,302 in the imputed data) the initial set of ~2,000 ClinVar P/LP variants found in QcP and gnomAD, rather than the ~200,000 P/LP variants reported in ClinVar. Furthermore, at the allele frequencies relevant here, an RFD of 10% corresponds to very small absolute differences that may be indistinguishable from sampling noise at the QcP and gnomAD sample sizes (see minor point #16 below). The RFD threshold is therefore confusing and difficult to justify, both as a preliminary filter before looking for founder variants, and for observations of mutation load beyond CR/IBD-defined founder variants, as in Figures 1 and 2. Given that the RFD threshold seems arbitrary to analyses of mutation load (the authors have explained that it would not affect their observations about load in SLSJ vs UQc), I would still suggest that these analyses be simply repeated on the entire set of P/LP variants, regardless of RFD, primarily for the sake of conceptual clarity. Otherwise, the authors should clearly explain in the manuscript why an initial RFD >= 10% threshold was necessary, as opposed to simply looking at all rare P/LP variants in QcP, citing for instance counts and MAFs of ClinVar P/LP variants observed in QcP before and after the filter.

ClinVar 1* vs 2* Review Status: The authors have explained that they did not repeat their analyses of mutation load in Figure 2 (or Figure 5) using only P/LP variants with review status >= 2*, because some 1* variants might be clinically relevant, including 11 of the previously-documented variants discussed in the study, and a newly-identified founder variant which was observed alongside the relevant phenotype. Because the aim of their study was to identify variants which might be linked to a founder disease, they did not find it necessary to restrict to >= 2* in any of their analyses. I agree that this rationale makes sense for identifying potentially pathogenic founder variants, however, another purpose of the study (included in its title) is to compare the mutation load of these variants between SLSJ and UQc. It would be informative and straightforward to add a simple sensitivity panel to Figs 2 and 5 restricted to ClinVar >=2* (plus previously documented founders) to show whether the trends persist when limited to variants with stronger evidence of pathogenicity. I also note that the term “mutation load” formally relates to fitness; if you keep this terminology, please define it operationally in terms of your potentially pathogenic count-based proxy and acknowledge limits (penetrance/inheritance/varying evidence of pathogenicity).

5. Reviewer assessment: Partially addressed. The authors have adequately clarified in the supplementary text that the additional 1,000 SLSJ individuals come from an independent cohort and are not a replication of CARTaGENE. The validation of the nine carrier rates is still referenced in lines 120-121; it would be good to very briefly add some clarifying wording here, e.g. “nine carrier rates were assessed in an independent cohort of 1,000 SLSJ individuals in the CIUSSS laboratory, and…” so that readers don’t need to go to the supplemental material to understand this.

Minor Points:

6. Reviewer assessment: Partially addressed. I appreciate the authors specifying they are concerned with rare Mendelian diseases. In following up on the edit made to line 60, however, I was unable to find any count of more than 10,000 rare Mendelian diseases in Orphanet, let alone rare Mendelian diseases. At the URL https://www.orpha.net/ as of July 2025 there are 6528 diseases under “Orphanet in numbers”, not all of which are necessarily genetic or Mendelian. The authors’ citation in the previous sentence (line 59, citation 1) is to Haendel et al. 2020, which estimates there are "more than 10,000" rare diseases (not necessarily all genetic or Mendelian), but this was using Mondo, which draws upon Orphanet and other resources like OMIM, rather than just within Orphanet. There is a count of Mendelian human diseases in Mondo at the URL https://mondo.monarchinitiative.org/ under “Representation of Disease Types”, which is at 11,566 for Mendelian human diseases as of July 2025. The authors should clarify this and add an appropriate citation.

7. Reviewer assessment: Adequately addressed. The authors have added the requested information to line 138.

8. Reviewer assessment: Adequately addressed. The authors have added a useful estimate that a recessive variant with a CR of 1/200 would result in approximately 7 affected individuals in SLSJ. I still think it would also be useful to include expected count/rate of affected individuals for all founder variants (not for a patient-focus, but because the frequency of a variant has different implications from a population-level depending on its inheritance pattern). However, because of the “already dense content of the tables” and the fact that inheritance patterns may not be readily available for all of the ClinVar P/LP variants reported in this manuscript, I accept that this information may be beyond the scope of the study.

9. Reviewer assessment: Adequately addressed. The authors have fixed the axis labels, and clarified the reason for variants with CR below 1/200 on lines 148-150.

10. Reviewer assessment: Adequately addressed. The authors have added gene names as requested.

11. Reviewer assessment: Adequately addressed. The authors have added phenotypes to what is now Table 2.

12. Reviewer assessment: Adequately addressed. The authors have added the other variant to each compound het case in Table 2.

13. Reviewer assessment: Adequately addressed. The authors have fixed the final count of genotyped samples.

14. Reviewer assessment: Adequately addressed. The authors have clarified in the Methods section that they did not use ClinVar variants with “Conflicting classifications of pathogenicity”.

15. Reviewer assessment: Adequately addressed. The authors have adequately shown in their response to this point that including close relatives (defined by sharing more than 1500cM total IBD) did not substantially affect estimates of carrier rates.

16. Reviewer assessment: Partially addressed. I appreciate the authors having added some conversions between RFD and fold-difference to the methods section. The authors use the RFD >= 10% threshold to indicate that variants are more common in QcP than in gnomAD v3.1.2 NFE genomes. There are 29,353 individuals in the imputed QcP dataset, and I believe there are something like ~33,000 individuals in the gnomAD v3.1.2 NFE genomes. At very low frequencies, an RFD of 10% corresponds to very small absolute differences (e.g., AF 0.005 vs 0.0045 at CR ~1/200). Would a 10% RFD for a variant with CR=1/200 in QcP be distinguishable from sampling noise for the QcP and gnomAD sample sizes? The authors should comment on this in the RFD Methods section.

17. Reviewer assessment: Adequately addressed. The authors have named the two MAF >= 0.05 variants which were removed, on lines 352-353 of the Methods section.

18. Reviewer assessment: Adequately addressed. The authors are correct that they had already addressed in the text (lines 365-366) why no lower than 5 individuals was used for an alternative observed carrier count threshold.

19. Reviewer assessment: Adequately addressed. The authors have clarified that the IBD pairwise sharing analyses were done separately for SLSJ, UQc, and QcP.

20. Reviewer assessment: Adequately addressed. The authors have provided the table in a readable format, and added gene names.

Reviewer #3: The authors addressed all of my comments.

**Have all data underlying the figures and results presented in the manuscript been provided?**

Reviewer #1: Yes

Reviewer #2: Yes

Reviewer #3: Yes

PLOS authors have the option to publish the peer review history of their article (what does this mean? ). If published, this will include your full peer review and any attached files.

**Do you want your identity to be public for this peer review?** For information about this choice, including consent withdrawal, please see our Privacy Policy .

Reviewer #1: No

Reviewer #2: No

Reviewer #3: No

**Figure resubmission:**
---

## [Editor Report · Decision Letter 2]

9 Sep 2025

Dear Dr Girard,

We are pleased to inform you that your manuscript entitled "Rare diseases load through the study of a regional population" has been editorially accepted for publication in PLOS Genetics. Congratulations!

Yours sincerely,

Jonathan Marchini

Academic Editor

PLOS Genetics

Gregory Cooper

Section Editor

PLOS Genetics

Aimée Dudley

Editor-in-Chief

PLOS Genetics

Anne Goriely

Editor-in-Chief

PLOS Genetics

Comments from the reviewers (if applicable):

**Data Deposition**

http://datadryad.org/submit?journalID=pgenetics&manu=PGENETICS-D-25-00284R2

**Press Queries**

---

## [Editor Report · Acceptance letter]

PGENETICS-D-25-00284R2

Rare diseases load through the study of a regional population

Dear Dr Girard,

We are pleased to inform you that your manuscript entitled "Rare diseases load through the study of a regional population" has been formally accepted for publication in PLOS Genetics! Your manuscript is now with our production department and you will be notified of the publication date in due course.

With kind regards,

Narmatha Raju, M.Sc

PLOS Genetics

On behalf of:
